# Digital Competence of Future Secondary School Teachers: Differences According to Gender, Age, and Branch of Knowledge

**David Jiménez-Hernández** *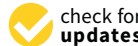, **Víctor González-Calatayud** *, **Ana Torres-Soto**,
**Asunción Martínez Mayoral** and **Javier Morales**

Center for Operational Research, Miguel Hernández University, 03202 Elche, Spain;
ana.torress@umh.es (A.T.-S.); asun.mayoral@umh.es (A.M.M.); javier.morales@umh.es (J.M.)
* Correspondence: djimenez@umh.es (D.J.-H.); victor.gonzalezc@umh.es (V.G.-C.)

**Abstract:** The development of related technological skills in secondary students is perceived as unachievable if the teachers do not have enough technological expertise to guide their students. This study was based on investigating digital competence on the population of graduate students undertaking a Master's degree in Education to train as teachers for the secondary educational level. The study made it possible to conclude, on the one hand, the homogeneity of university degrees within the scope of the Bologna Plan, with respect to mean levels of digital training. On the other hand, more exhaustively, differences have come to light concerning specific training on the different digital competence areas in the DigComp evaluation system, related to gender, branch of knowledge, age, and by considering the self-perception of individuals on their own capabilities in everyday technological issues. Consequently, the need for incorporating an ICT syllabus into the subjects covered by a Master's in education has been highlighted, as well as promoting females, older people, and nontechnological degrees. Digital training on critical competence areas for teaching, such as the creation of digital content, suffers from major shortcomings and should also be promoted.

**Keywords:** digital competence; teacher professional development; secondary education; technological skills

## 1. Introduction

It is a fact that 21st-century technology has invaded our lives. Although quite a lot has been incorporated into education, there is still a long way to go. In order to do so, and within the final framework we study, related to secondary education in Spain (between 12 and 17 years old), it is absolutely necessary to develop what is known as digital competence, both for teachers and students.

Digital competence can be understood as those values, beliefs, knowledge, skills, and attitudes to adequately use technologies, including both computers and the different telematic programs and tools that allow and enable the search, access, organization, and use of information, to adequately respond to the different demands of the environment and build knowledge [1]. Specifically, in the laws that regulate the Spanish education system until the age of 17 [2], this competence is understood as that which implies the creative, critical, and safe use of information and communication technologies, to achieve objectives related to work, employability, learning, use of free time, inclusion, and participation in society.

In the Spanish higher education system, based on the framework established by the European Union [3], a series of basic competencies are included in the law that regulate our system, which should be acquired by all students by the end of their university education [4]. Among these competencies,

those related to information processing and the digital ones stand out. These ones might be the premises for all teachers embarking on teaching at the secondary education level.

The development of related technological skills in secondary students is perceived as unachievable if the teachers do not have enough technological expertise to guide their students [5]. In Spain, graduates must complete a one-year master's degree that qualifies them for the teaching profession. Reference [6] pointed out the importance of an "initial teacher training on ICT", since it allows teachers to join the educational system with an adequate level of digital competence.

Our interest focused on investigating the initial level of technological skills of future teachers entering the mentioned master's program in Spain, as achieved by completing their undergraduate studies at the university under the Bologna digital competence paradigm. Conclusions will be the starting point to define specific needs in the initial teacher training, as defined in the syllabus of specific subjects in ICT and didactic uses of technology in several of the master's degrees that qualify for the teaching profession in Spain.

## 2. Digital Competence and Associated Factors

Evaluation of digital competence (DC) has been a challenge for years. At present, we count on several procedures and rubrics that facilitate a systematized evaluation. Moreover, three factors have been historically related to digital competence: gender, age, and branch of knowledge. All these factors are revised and presented below.

### 2.1. Frameworks for DC and Its Assessment

Several national and international reference works have been developed to define digital competence and delimit its components. Table 1 shows the main models and their dimensions or areas.

**Table 1.** Main models for studying digital competence, and areas of teachers' digital competence (TDC) [7].

| Model Framework | Institution | Reference | Areas or Dimensions of TDC |
|---|---|---|---|
| NETS-T | ISTE | ISTE (2008) | Learning and creativity of the students, learning and evaluation experiences, work, citizenship, and professional growth. |
| Teachers ICT competence standards | UNESCO | Unesco (2008) | Policy and vision, curriculum and evaluation, pedagogy, ICT, organization and administration, professional teacher training. |
| DigiLit Leicester | Leicester City Council | Fraser, Atkins & Richard (2013) | Search, evaluation, and organization; create and share; evaluation and feedback; communication; collaboration; and participation, security, identity, and development. |
| DIGCOMP-EDU | European Commission | Redecker & Punie (2017) | Social and professional commitment, digital resources, digital pedagogy, evaluation and feedback, empowerment of students, facilitating students' digital competence. |
| Common Framework for TDC | Ministry of Education, Government of Spain | INTEF (2014 y 2017) | Information, communication, content creation, security, problem solving. |

These different conceptions of DC have led to the creation of different assessment tools in relation to these models. In the work of [8], the main instruments created to date are included. Those instruments that have focused on the DIGCOMP model [9] stand out, fundamentally because it is the base for the established model developed in Spain for teachers [10]. This model is the basis of our work, for which we have developed a simplified questionnaire, detailed in Section 3.2, to evaluate the five areas of competence considered there.

Various works have been based on finding differences in digital competence within the teacher population of Spain. Ref. [11] found that the area in which the teachers in initial training showed the highest level was the area of information and the lowest one was content creation. However, in the work of [12], a lower level was found in the area of problem solving.

## 2.2. Relationship Between Digital Competence and Gender

Gender-based digital competence and the digital divide have been extensively studied, with controversial results. The study carried out by the OECD (Organization for Economic Cooperation and Development) and compiled by [13], which studied the factors that influence the digital divide, showed how the divide fluctuates depending on the country, without a clearly marked pattern.

In the case of university students in Spain, although digital competence is a transversal element in training, included in the Bologna Plan, there is no agreement on its uniformity between genders. Reference [14] found differences in ICT skills related to gender in three of the seven variables studied, with higher scores for women. In contrast, Reference [15], found that students of the Degree in Physical Activity and Sport shared similar technological knowledge and didactic use of technology; however, they found gender differences in relation to digital competence, favoring males. Reference [16] found no clear differences in digital competence favoring one gender over the other in undergraduates training to be teachers.

If we focus on the gender differences in teachers, there are still discrepancies. On the one hand, we found research where a clear difference was found, as is the case in [17], which showed that male teachers had a greater capacity to gather information by technological means than female teachers. Or the study by [18], in which gender was a more relevant variable than age for understanding differences in knowledge of technological resources. Likewise, in two studies carried out in Chilean universities with teachers, differences in digital competence were found, with men scoring higher than women [19,20].

On the other hand, we found a number of studies where the difference between men and women was not so clear. This was the case in the study by [21], which found differences in ICT competence in favor of men, but these gradually disappeared. In addition, women scored higher in the dimension of integration integrating ICT into their teaching practice. In another study with teachers, Reference [22] found differences between genders in the communication and collaboration variables (with women scoring higher than men), and content creation and security (with men scoring the highest).

Others, such as [23], observed that the variables of gender, age, experience, type of school, stage, or area, did not influence the development of digital teaching skills. Finally, the studies by [24,25], did not find differences by gender relating to the digital competence of teachers either.

As it has become clear, there is no consensus on a gender divide in digital competence within the population of teachers or future teachers in Spain.

## 2.3. Digital Competence According to Age

Digital competence and its association with the age variable has been under study for some time. In 2001, Marc Prensky made a generational distinction, naming those who were born in the ICT era (those born after 90) as digital natives and the rest as immigrants. Other authors have called these natives the "Z-Generation". With this generational differentiation, it is assumed that these digital natives have skills that immigrants do not have in the use of technologies.

Nowadays, most of our graduate students training to become teachers at secondary level through the Master's in Secondary Education, come from this so-called Z-generation group; hence the interest in investigating differences in basic (even medium or advanced) digital skills acquisition before starting this master's degree.

Recent studies suggested that this new generation of digital natives are not as competent as they are supposed to be [26,27]. These works showed that these students, although capable of using technology in their day-to-day lives for communication and leisure, they did not use it as proficiently or even routinely for academic learning.

Reference [28] found associations between the age and attitude dimensions of students, in favor of the younger ones. Along the same lines, Reference [29] identified a higher perception of digital competence in the youngest group under study (between 20 and 24 years old). In contrast, other studies with university students showed no positive correlation between age and digital competence [30–32].

Concerning the digital competence of teachers, there is a clear trend in identifying that younger teachers are considered more technologically competent than older ones. Reference [21] detected an overall trend linking older teachers to poorer mastery of technological resources, even in those relating to day-to-day use. Likewise, the study undertaken by [24] found significant differences according to age, corroborating a much higher teacher training profile in ICT in younger or less experienced teachers. The recent study carried out by [33] also showed that younger teachers performed better in content creation and problem-solving, and evidenced an inverse relationship between age and digital competence. However, the study carried out by [23] found no association between age and the development of digital teaching competence.

## 2.4. Digital Competence According to the Branch of Knowledge

The Spanish university system organizes its studies in five branches of knowledge: arts and humanities, sciences, health sciences, social and legal sciences, and engineering and architecture. The branch of knowledge is another variable associated with the development of digital competence. Even with the incorporation of competence skills in the Bologna academic plans, Reference [14] found significant differences in digital competence according to this variable, though not for all its dimensions. With regard to teachers' digital competence, Reference [24] stated that teachers in primary and secondary school with a scientific-technological profile displayed better digital competences than those coming from other branches. Reference [23], on the other hand, did not find any differences related to the areas they taught, except for teachers in the areas of arts and crafts, music, or technology, who proved to be more qualified in using certain technological tools.

## 3. Materials and Methods

The main objective of our research was to analyze the digital competence of future secondary school teachers in Spain (those teaching young people between 12 and 17 years old), at the time they started, as graduates, the Master in Education to initiate their specific teacher training. Relationships between digital competence and gender, age, and branch of knowledge were investigated. Specifically, the proposed goals, within the Spanish framework of students starting this master's degree, were:

- Goal 1: to investigate the impact of Bologna guides on university degrees, for a homogeneous training in digital standards.
- Goal 2: to estimate the effect of gender, age, and branch of knowledge to predict digital competence in each of the five areas of competence proposed in DigComp.

### 3.1. Procedure

Data collection was carried out during the first month of classes in one of the Master's in Education courses in Spain (detailed below), for three consecutive academic years (2017–18 to 2019–20), by filling in a voluntary online questionnaire passed on to students beginning the master's. The research was carried out following all the ethical principles established by the Office of Responsible Research of the University Miguel Hernández of Elche.

### 3.2. Instruments

A simplified questionnaire based on the DigComp model was used. It included the five areas of competence, say: A1, information and informational literacy (3 items); A2, communication and collaboration (6 items); A3, digital content creation (4 items); A4, security (4 items); and A5, problem solving (4 items). All items were scaled from 1 to 5, 1 being the minimum level of competence acquisition and 5 the maximum.

Reliability of the questionnaire at the area level has been studied through Cronbach's alpha coefficient for each group of items defining the competence areas A1 to A5. Resulting coefficients

and 95% confidence intervals are shown in Table 2. All of them provide acceptable results in favor of reliability of the questionnaire for all areas of competence (coefficients above 0.75).

**Table 2.** Cronbach's alpha coefficient and 95% confidence intervals for the five groups of items defining the five areas of competence A1 to A5.

|  | **A1** | **A2** | **A3** | **A4** | **A5** |
|---|---|---|---|---|---|
| Alpha coef. (95% conf.int) | 0.79 (0.77,0.81) | 0.82 (0.81,0.83) | 0.76 (074,0.78) | 0.83 (0.81,0.84) | 0.82 (0.81,0.83) |

Variables of interest collected from the survey, and considered in the analysis, were:

- Year of birth, categorized into those born before and since 1990. This variable is referred to as ZG, with levels 0/1 and 1 identifying Z-Generation individuals.
- Gender (male/female).
- Branch of knowledge (BK) of the degree with which the student reaches the master, with levels arts and humanities (AH), sciences (S), health sciences (HS), social and legal sciences (SLS), and engineering and architecture (EA).
- Self-perception of everyday technological skills (ETS), collected on the basis of an integer 1 to 5 scale question. The original response was categorized into three skill levels: min = levels 1 and 2, med = levels 3 and 4, max = level 5.
- Competence area (CA) identifies each one the five areas of digital competence, with levels A1 to A5.
- Digital competence area (DCA) contains the mean values assessed for each competence area, derived by averaging the concerned item responses on the individual self-perceptions. It gives the digital competence score.
- For each individual, a grand mean was assessed to evaluate the global digital competence level (GDC). Individual responses to all 25 items in the questionnaire were excluded from the analysis once the area and global means had been assessed.
- An identification variable ID was also created for each subject, which allowed us to include all available information in the modelling and link the individual responses, by preserving the independence between different subjects.

### 3.3. Participants

This study focused on investigating the digital competence of future teachers embarking on the Master in Education, to initiate specific teacher training at the secondary education level in Spain. At XXX (Spain), where this study was based, this master's degree is called Master in Secondary Education Teaching, Vocational Training, and Language Teaching (referred to as MasterProf from now on). Due to the dimensions of this master's, with more than 400 students each academic year, it could be considered representative of any other Master in Education in Spain, and so their students as representative of the total population of interest. Moreover, the available sample of 485 individuals, covered the 2017–2018, 2018–2019, and 2019–2020 academic years, thereby providing a more dynamic view over time. The available sample included 40.4% of the total enrolment for all three years of the MasterProf program, so it can be considered as representative sample of those who have completed the master's degree in question. The sample was selected for convenience, asking those enrolled in the master's to answer the questionnaire anonymously.

Next, a description of the available data is presented. The total number of respondents, 485, is presented in Tables 3–5, respectively, classified by course and age (Z-generation), gender, and everyday technological skills perceived, and the branch of knowledge (AH = arts and humanities, S = science, HS = health science, SLS = social science and law, EA = engineering and architecture).

**Table 3.** Number of respondents by academic year and age (given by belonging to the Z-generation).

| Academic Year 2017–2018 | | Academic Year 2018–2019 | | Academic Year 2019–2020 | |
|---|---|---|---|---|---|
| **Born before 1990** | **Born since 1990 (Z-Gen)** | **Born before 1990** | **Born since 1990 (Z-Gen)** | **Born before 1990** | **Born since 1990 (Z-Gen)** |
| 80 | 87 | 61 | 62 | 77 | 118 |

**Table 4.** Number of respondents by gender and everyday technological skills self-perception.

| Everyday Technological Skills (ETS) | | | | | |
|---|---|---|---|---|---|
| **Min** | | **Med** | | **Max** | |
| **Male** | **Female** | **Male** | **Female** | **Male** | **Female** |
| 12 | 26 | 54 | 99 | 121 | 173 |

**Table 5.** Number of respondents by branch of knowledge (BK) and gender.

| Branch of Knowledge (BK) | | | | | | | | | |
|---|---|---|---|---|---|---|---|---|---|
| **AH** | | **S** | | **HS** | | **SLS** | | **EA** | |
| **Male** | **Female** | **Male** | **Female** | **Male** | **Female** | **Male** | **Female** | **Male** | **Female** |
| 54 | 92 | 22 | 50 | 3 | 32 | 66 | 80 | 42 | 44 |

Note: AH = arts and humanities, S = science, HS = health science, SLS = social science and law, EA = engineering and architecture.

All data gathered from the three academic years were pooled together in the analysis. In fact, the ANOVA test for differences in the global competence indicated nonsignificant results (*p*-value of 0.526). Lacks of normality for the global means were not evidenced (Shapiro-test for normality gave a *p*-value of 0.344); this result validated the ANOVA test for the difference in means.

As the number of respondents in the health science branch (HS) was so small (especially for males, with just 3 individuals), the HS and science (S) branches were grouped together in S. The difference in the means of global competence for these two groups was not significant (2.78 ± 0.56 for S and 2.75 ± 0.8 for HS; *p*-value of 0.805 for the *t*-test). The means and standard deviations (in brackets) calculated for each of the competence areas, and also globally, are shown in Table 6.

**Table 6.** Means and standard deviations (in brackets) of digital competence indicators for the five areas, and also for the global digital competence.

| | | Digital Competence Area (DCA) | | | | | |
|---|---|---|---|---|---|---|---|
| | | **A1** | **A2** | **A3** | **A4** | **A5** | **Global Mean (GDC)** |
| **Gender** | **Male** | 3.51(0.83) | 3.25(0.77) | 2.29(0.86) | 3.03(0.87) | 2.99(0.88) | 3.01(0.72) |
| | **Female** | 3.31(0.79) | 3.17(0.81) | 2.14(0.81) | 2.9(0.88) | 2.72(0.85) | 2.85(0.68) |
| **BK (Branch of Knowledge)** | **AH** | 3.34(0.9) | 3.2(0.86) | 2.24(0.83) | 2.99(0.88) | 2.8(0.94) | 2.92(0.75) |
| | **CC** | 3.31(0.74) | 3.06(0.77) | 2.03(0.71) | 2.83(0.87) | 2.65(0.85) | 2.78(0.65) |
| | **CSJ** | 3.33(0.8) | 3.18(0.78) | 2.14(0.85) | 2.9(0.87) | 2.76(0.83) | 2.86(0.69) |
| | **INA** | 3.66(0.69) | 3.39(0.72) | 2.45(0.89) | 3.11(0.89) | 3.18(0.74) | 3.16(0.64) |
| **ETS (Everyday Tech Skills)** | **Min** | 2.24(0.85) | 2.07(0.71) | 1.32(0.6) | 1.91(0.82) | 1.66(0.74) | 1.84(0.61) |
| | **Med** | 3.07(0.66) | 2.89(0.65) | 1.9(0.65) | 2.67(0.8) | 2.44(0.65) | 2.59(0.49) |
| | **Max** | 3.7(0.67) | 3.51(0.68) | 2.47(0.81) | 3.23(0.77) | 3.17(0.76) | 3.22(0.59) |
| **ZG (Z-Generation)** | **Before 90** | 3.26(0.89) | 3.04(0.86) | 2.03(0.84) | 2.81(0.94) | 2.7(0.89) | 2.77(0.77) |
| | **Since 90** | 3.5(0.73) | 3.33(0.72) | 2.34(0.8) | 3.06(0.81) | 2.92(0.84) | 3.03(0.62) |

*3.4. Data Analysis*

The main objectives of the study were to investigate the possible conditioning effects on digital competence of the following variables: gender, age, and branch of knowledge, but also including self-perception of everyday digital skills. Normal linear Bayesian mixed models, with fixed and random effects, were proposed to identify relevant effects, both studying the global digital competence and also the individual area skills. Normality was tested on the residuals of the fitted models, in order to validate the results.

Model fitting was done from a Bayesian perspective to answer both goals. When studying digital competence by areas, random effects are included in the model to link responses from the same individual and to take advantage of all available information on every subject. From a Bayesian perspective, random effects can be interpreted in a more intuitive way, as intrinsic variation features of the individuals in the sample, not affecting the conclusions on the conditioning variables. To answer goal one, no random effects were necessary, but benefits of Bayesian interpretations were exploited.

The identification of relevant factors in explaining the response was resolved through a variable selection procedure starting from a full model with pairwise interactions. Selection criteria used were: marginal likelihood (MLIK) [34]; deviance information criterion (DIC), proposed by [35]; and effective number of parameters (ENP), see [36], which provides a measure of the complexity of a model. Models with lower values in DIC and ENP and maximum MLIK are preferred. In the backward selection procedure, a variable/effect is a candidate for exiting the model if at least two of the three proposed selection criteria provide an improvement.

Inferences and conclusions are given in terms of posterior means (as estimates), standard deviations (as error measures for the estimates), and credible intervals with probability 0.95 for coefficients/effect sizes and/or for group means. Given a 95% credible interval (a,b) for an effect size m, "a" represents the value for which the probability encountered for the effect size "m" to be greater than "a" is 0.975, and the probability for the effect size "m" to be between "a" and "b" is 0.95. Posterior inferences combine minimum prior information (in this case), and that provided by the available data.

The statistical analysis was solved with software R and RStudio, and packages tidyverse [37] and INLA developed by [38–40], among others.

## 4. Results

The results shown below have been divided according to the objectives set out in the research. Firstly, global results are provided about the impact of a theoretical homogeneous training in digital standards over diverse university degrees in Spain, coming from different branches of knowledge. Secondly, differences among digital competence area achievements were studied in terms of the variables of interest, say: gender, age, and branch of knowledge, and also the personal perception about his/her digital performance in everyday issues.

*4.1. Digital Homogeneity Training on Bologna Degrees*

Firstly, we investigated the homogeneity in digital training among the university degrees in Spain represented in the sample coming from the Bologna system, as classified in the different branches of knowledge. We therefore restricted the study to sample data of individuals born after 1991 and therefore who started their undergraduate studies from 2009–2010, when the Bologna system was fully implemented in Spain. The sample size of this restricted sample was 234 individuals and 102 different degrees in the four collapsed branches of knowledge (AH, S, HS, EA).

Though variables BK, gender, and ETS were initially included in the modelling (as well as their interactions), the resulting model remained:

$$\text{GDC} \sim 1+ \text{GENDER} + \text{ETS}. \tag{1}$$

This fact implies that digital competence is clearly linked to self-perception about everyday tech skills, but also to gender differences. However, the branch of knowledge to which the degrees were subscribed to, did not contribute to explain statistical differences on digital competence from a global perspective.

The global digital competence was −0.108 smaller for females than males (95% credible interval (−0.245, 0.029)). Moreover, the smaller the perception of tech skills on everyday routines, the smaller the global digital competence as assessed by the survey questionnaire (which proved consistent): the mean difference between med and min ETS levels was 0.288, and its (posterior) probability of being greater than −0.022 was 0.975. The difference between max and min ETS levels was 0.931, with a probability of 0.975 of being greater than 0.636 (as given by the 95% credible interval). Nonexplained variance by the model, that is, the variance of the residuals, was considerably small 0.272, as was its 95% credible interval (0.228, 0.324), which lends reliability to the results.

In Figure 1, the posterior means and credible intervals (in brackets) are displayed for the different population groups defined by the conditioning factors, say gender (males and females, and ETS (minimum, medium, and maximum). Differences among groups are clearly identified.

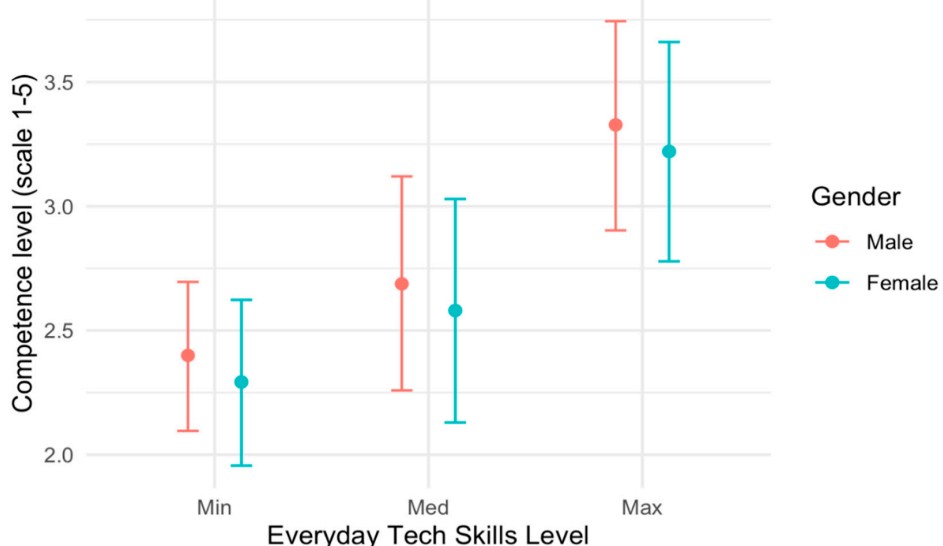

**Figure 1.** Posterior estimates of the global digital competence, based on gender and self-perception of everyday technological skills.

### 4.2. Specific Digital Skills and Conditioning Factors

The proposed starting linear model to predict the digital score in all competence areas, contained in the DCAS variable, is given by the formula:

$$DCAS \sim 1 + CA + GENDER + BK + ETS + ZG + \tag{2}$$

$$CA{:}GENDER + CA{:}BK + CA{:}ETS + CA{:}ZG + \tag{3}$$

$$GENDER{:}BK + GENDER{:}ETS + GENDER{:}ZG + \tag{4}$$

$$BK{:}ETS + BK{:}ZG + \tag{5}$$

$$ETS{:}ZG + Rand.Effect(ID,iid) \tag{6}$$

where fixed effects and pairwise interactions are represented in capital letters, and random effects linking all data from the same individual in "Rand.Effect(ID,iid)". This random effect assumes a priori independence between individuals and also among the means of the five competence area scores for

each individual, so as not to lead to erroneous assumptions that a subject who scores high in one area will necessarily score high in other areas.

The resulting model after variable selection is given by:

$$DCAS \sim 1 + CA + GENDER + BK + ETS + ZG +$$
$$+ CA{:}ETS + ETS{:}ZG + Rand.Effect(ID,iid), \tag{7}$$

where all main effects remain and also the interactions CA with ETS (which explains if the areas that score the most are or are not the same, depending on the everyday technological skills of subjects) and ETS with ZG (which reflects possible differences in ETS performance for individuals born after or before 1990).

The estimated variance for the random effects (identifying individuals) was 0.257, with 95% credible interval (0.24, 0.27). This result endorses the need to link individual observations.

The coefficient estimates of the fitted model are shown in Table 7. Relevant facts derived from the fit are:

- Females scored below males by 0.11 points in the 1–5 scale.
- Competence area A3 was the one that scored the lowest for all individuals, related with digital content generation, followed by A5, A4, A2, and A1. Digital competence, A1, the basic one and more related with everyday tech skills, was the most developed digital competence for all individuals.
- Competence area estimated scores did not behave the same for all groups of individuals as classified by their everyday tech skills (ETS); the interaction effect CA:ETS was relevant.
- The branch of knowledge (BK effect) for which a lower score was predicted for all the competence areas was science (S), followed by social and legal sciences (SLS) and then arts and humanities (AH). The one that scored the highest was engineering and architecture (EA).
- Individuals born from 1990 onwards (Z-Generation), scored higher in all competence areas (ZG effect).
- The effect of Z-Generation (ZG) interacting with ETS was remarkable. This effect is shown in Figure 2, where predictions for male individuals in BK = sciences are displayed (their behavior was common for all branches and genders). Differences between Z-Generation and those born before 1990 were more severe for the group that perceived itself as possessing minimal technological skills. Differences were small and minimum for the groups in maximum and medium ETS, respectively.

**Table 7.** Posterior estimates of model coefficients in the fitted model for digital competence score by area: means and 95% credible intervals.

| Coefficients | Mean (95% CI) | Coefficients | mean (95% CI) |
|---|---|---|---|
| (Intercept) | 2.11(1.83,2.39) | ZG | 0.70(0.33,01.06) |
| CA_A2 | −0.16(−0.39,0.07) | CA_A2:ETS_Med | −0.02(−0.28,0.23) |
| CA_A3 | −0.91(−1.14,−0.69) | CA_A3:ETS_Med | −0.26(−0.52,−0.01) |
| CA_A4 | −0.33(−0.56,−0.10) | CA_A4:ETS_Med | −0.07(−0.33,0.18) |
| CA_A5 | −0.58(−0.81,−0.35) | CA_A5:ETS_Med | −0.06(−0.31,0.20) |
| GENDER_Female | −0.11(−0.22,−0.01) | CA_A2:ETS_Max | −0.03(−0.28,0.21) |
| BK_S | −0.11(−0.25,0.02) | CA_A3:ETS_Max | −0.32(−0.56,−0.07) |
| BK_SLS | −0.09(−0.22,0.04) | CA_A4:ETS_Max | −0.14(−0.39,0.10) |
| BK_EA | 0.09(−0.07,0.24) | CA_A5:ETS_Max | 0.05(−0.19,0.29) |
| ETS_Med | 1.07(0.77,1.37) | ETS_Med:ZG | −0.66(−1.06,−0.26) |
| ETS_Max | 1.60(1.31,1.89) | ETS_Max:ZG | −0.55(−0.94,−0.16) |

Numerical values for the predictions of the digital scores in Figure 2 are also displayed in Tables 8–10, which show the posterior estimates of mean scores in the five digital competence areas, differentiated by gender, branch of knowledge (BK), and age (born before and since 1990). Each one

of the three tables identifies each one of the levels of the self-perceived everyday tech skills variable. The 95% credible intervals for predictions are displayed in brackets.

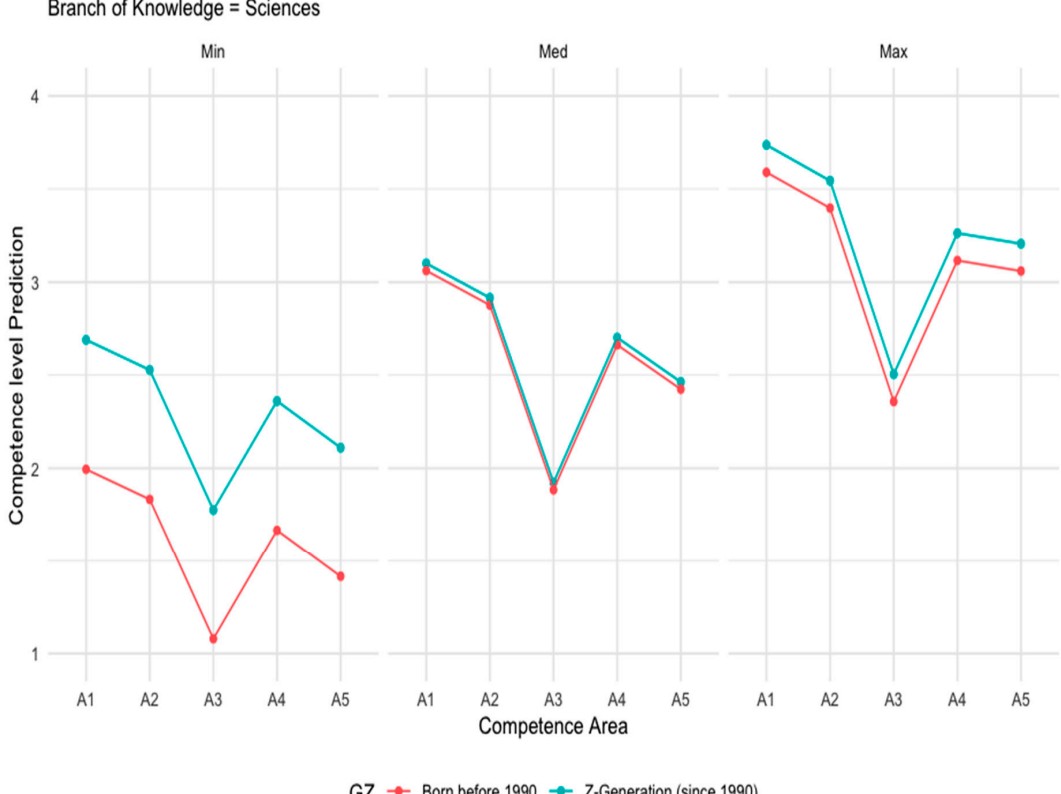

**Figure 2.** Prediction of competence score by competence area, ETS, and GZ. Behavior for male subjects in the Science Branch of Knowledge.

**Table 8.** Predictions of digital achievement in every competence area for minimum ETS. The 95% credible intervals are in brackets.

| ETS = Minimum | | | | | | |
|---|---|---|---|---|---|---|
| **Born Before 1990** | | **Digital Competence Area (DCA)** | | | | |
| **GENDER** | **BK** | **A1** | **A2** | **A3** | **A4** | **A5** |
| Male | AH | 2.11(1.82,2.38) | 1.95(1.58,2.31) | 1.19(0.83,1.56) | 1.78(1.41,2.14) | 1.53(1.17,1.89) |
| Female | AH | 2(1.7,2.29) | 1.83(1.45,2.21) | 1.08(0.71,1.46) | 1.67(1.29,2.04) | 1.42(1.04,1.79) |
| Male | S | 1.99(1.68,2.3) | 1.83(1.44,2.22) | 1.08(0.69,1.47) | 1.67(1.28,2.05) | 1.41(1.03,1.8) |
| Female | S | 1.88(1.55,2.21) | 1.72(1.32,2.12) | 0.97(0.57,1.37) | 1.55(1.15,1.95) | 1.3(0.9,1.7) |
| Male | SLS | 2.01(1.7,2.32) | 1.85(1.47,2.23) | 1.1(0.71,1.48) | 1.69(1.3,2.07) | 1.44(1.05,1.82) |
| Female | SLS | 1.9(1.58,2.22) | 1.74(1.34,2.14) | 0.99(0.59,1.38) | 1.57(1.17,1.98) | 1.32(0.92,1.72) |
| Male | EA | 2.2(1.87,2.52) | 2.03(1.64,2.43) | 1.28(0.89,1.68) | 1.87(1.47,2.26) | 1.62(1.22,2.01) |
| Female | EA | 2.08(1.74,2.42) | 1.92(1.51,2.33) | 1.17(0.77,1.58) | 1.75(1.34,2.16) | 1.5(1.09,1.91) |
| **Born since 1990 (Z-Generation)** | | **Digital Competence Area (DCA)** | | | | |
| **GENDER** | **BK** | **A1** | **A2** | **A3** | **A4** | **A5** |
| Female | AH | 2.69(2.23,3.16) | 2.53(2,3.05) | 1.78(1.26,2.3) | 2.36(1.84,2.9) | 2.11(1.6,2.63) |
| Male | S | 2.69(2.21,3.17) | 2.53(2,3.06) | 1.78(1.25,2.31) | 2.36(1.83,2.9) | 2.11(1.59,2.64) |
| Female | S | 2.58(2.09,3.07) | 2.42(1.87,2.96) | 1.66(1.13,2.2) | 2.25(1.71,2.79) | 2(1.47,2.53) |
| Male | SLS | 2.71(2.25,3.18) | 2.55(2.03,3.07) | 1.8(1.27,2.32) | 2.38(1.86,2.92) | 2.13(1.62,2.66) |
| Female | SLS | 2.6(2.12,3.08) | 2.44(1.91,2.97) | 1.68(1.15,2.22) | 2.27(1.73,2.82) | 2.02(1.49,2.56) |
| Male | EA | 2.89(2.41,3.38) | 2.73(2.19,3.27) | 1.98(1.45,2.51) | 2.56(2.03,3.11) | 2.31(1.79,2.85) |
| Female | EA | 2.78(2.29,3.27) | 2.62(2.07,3.16) | 1.86(1.32,2.41) | 2.45(1.9,3) | 2.2(1.67,2.75) |

**Table 9.** Predictions of digital achievement in every competence area for medium ETS. The 95% credible intervals are in brackets.

| | | ETS = Medium | | | | |
|---|---|---|---|---|---|---|
| **Born Before 1990** | | **Digital Competence Area (DCA)** | | | | |
| **GENDER** | **BK** | **A1** | **A2** | **A3** | **A4** | **A5** |
| Male | AH | 3.18(2.75,3.58) | 2.99(2.45,3.53) | 2(1.45,2.53) | 2.78(2.23,3.32) | 2.54(2.01,3.07) |
| Female | AH | 3.06(2.63,3.48) | 2.88(2.32,3.43) | 1.89(1.33,2.43) | 2.66(2.11,3.21) | 2.43(1.89,2.98) |
| Male | S | 3.06(2.62,3.5) | 2.88(2.32,3.44) | 1.88(1.32,2.44) | 2.66(2.1,3.22) | 2.42(1.89,2.99) |
| Female | S | 2.95(2.5,3.4) | 2.76(2.2,3.33) | 1.77(1.2,2.34) | 2.55(1.99,3.11) | 2.31(1.76,2.88) |
| Male | LSL | 3.08(2.65,3.51) | 2.9(2.34,3.45) | 1.9(1.35,2.45) | 2.68(2.13,3.24) | 2.45(1.91,3) |
| Female | LSL | 2.97(2.52,3.41) | 2.78(2.22,3.34) | 1.79(1.23,2.35) | 2.57(2.01,3.13) | 2.33(1.78,2.9) |
| Male | EA | 3.26(2.82,3.7) | 3.08(2.51,3.64) | 2.09(1.52,2.65) | 2.86(2.3,3.42) | 2.63(2.08,3.19) |
| Female | EA | 3.15(2.69,3.6) | 2.97(2.4,3.54) | 1.97(1.4,2.54) | 2.75(2.18,3.32) | 2.51(1.95,3.08) |
| **Born since 1990 (Z-Generation)** | | **Digital Competence Area (DCA)** | | | | |
| **GENDER** | **BK** | **A1** | **A2** | **A3** | **A4** | **A5** |
| Male | AH | 3.21(2.54,3.91) | 3.03(2.27,3.8) | 2.04(1.27,2.79) | 2.82(2.05,3.59) | 2.58(1.83,3.33) |
| Female | AH | 3.1(2.42,3.8) | 2.92(2.15,3.69) | 1.92(1.15,2.69) | 2.7(1.95,3.48) | 2.46(1.71,3.23) |
| Male | S | 3.1(2.42,3.79) | 2.92(2.14,3.69) | 1.92(1.15,2.7) | 2.7(1.94,3.48) | 2.46(1.7,3.23) |
| Female | S | 2.99(2.3,3.69) | 2.8(2.02,3.59) | 1.81(1.04,2.6) | 2.59(1.82,3.37) | 2.35(1.59,3.13) |
| Male | LSL | 3.12(2.44,3.82) | 2.94(2.16,3.71) | 1.94(1.16,2.72) | 2.72(1.95,3.51) | 2.48(1.72,3.25) |
| Female | LSL | 3.01(2.33,3.72) | 2.82(2.04,3.61) | 1.83(1.04,2.61) | 2.61(1.83,3.4) | 2.37(1.6,3.14) |
| Male | EA | 3.3(2.62,4) | 3.12(2.35,3.9) | 2.12(1.34,2.91) | 2.9(2.13,3.69) | 2.67(1.89,3.42) |
| Female | EA | 3.19(2.5,3.9) | 3(2.23,3.78) | 2.01(1.23,2.8) | 2.79(2.01,3.58) | 2.55(1.78,3.33) |

**Table 10.** Predictions of digital achievement in every competence area for maximum ETS. The 95% credible intervals are in brackets.

| | | ETS = Maximum | | | | |
|---|---|---|---|---|---|---|
| **Born before 1990** | | **Digital Competence Area (DCA)** | | | | |
| **GENDER** | **BK** | **A1** | **A2** | **A3** | **A4** | **A5** |
| Male | AH | 3.7(3.29,4.12) | 3.51(2.99,4.03) | 2.47(1.94,2.99) | 3.23(2.71,3.75) | 3.17(2.64,3.7) |
| Female | AH | 3.59(3.17,4.01) | 3.4(2.86,3.94) | 2.36(1.81,2.89) | 3.12(2.58,3.66) | 3.06(2.53,3.6) |
| Male | S | 3.59(3.16,4.02) | 3.4(2.86,3.94) | 2.36(1.81,2.89) | 3.12(2.57,3.66) | 3.06(2.51,3.6) |
| Female | S | 3.48(3.03,3.92) | 3.28(2.73,3.84) | 2.25(1.68,2.79) | 3(2.45,3.55) | 2.95(2.39,3.49) |
| Male | SLS | 3.61(3.18,4.05) | 3.42(2.88,3.95) | 2.38(1.83,2.91) | 3.14(2.6,3.68) | 3.08(2.54,3.62) |
| Female | SLS | 3.5(3.06,3.94) | 3.31(2.75,3.86) | 2.27(1.71,2.8) | 3.02(2.48,3.58) | 2.97(2.42,3.51) |
| Male | EA | 3.79(3.35,4.24) | 3.6(3.06,4.15) | 2.56(2.01,3.1) | 3.32(2.77,3.87) | 3.26(2.71,3.81) |
| Female | EA | 3.68(3.23,4.13) | 3.49(2.93,4.04) | 2.45(1.89,3) | 3.21(2.65,3.76) | 3.15(2.59,3.71) |
| **Born since 1990 (Z-Generation)** | | **Digital Competence Area (DCA)** | | | | |
| **GENDER** | **BK** | **A1** | **A2** | **A3** | **A4** | **A5** |
| Male | AH | 3.85(3.18,4.53) | 3.66(2.92,4.41) | 2.62(1.88,3.36) | 3.38(2.64,4.14) | 3.32(2.58,4.07) |
| Female | AH | 3.74(3.06,4.42) | 3.55(2.8,4.3) | 2.51(1.75,3.25) | 3.26(2.52,4.03) | 3.21(2.45,3.97) |
| Male | S | 3.74(3.05,4.43) | 3.54(2.79,4.3) | 2.51(1.75,3.26) | 3.26(2.52,4.04) | 3.21(2.46,3.96) |
| Female | S | 3.62(2.93,4.33) | 3.43(2.67,4.21) | 2.39(1.63,3.15) | 3.15(2.39,3.94) | 3.09(2.34,3.87) |
| Male | SLS | 3.76(3.08,4.45) | 3.57(2.82,4.33) | 2.53(1.77,3.28) | 3.28(2.54,4.05) | 3.23(2.49,3.98) |
| Female | SLS | 3.64(2.97,4.34) | 3.45(2.7,4.22) | 2.41(1.65,3.17) | 3.17(2.42,3.95) | 3.11(2.36,3.88) |
| Male | EA | 3.94(3.26,4.62) | 3.75(2.99,4.5) | 2.71(1.94,3.47) | 3.47(2.71,4.24) | 3.41(2.66,4.18) |
| Female | EA | 3.83(3.13,4.52) | 3.63(2.86,4.4) | 2.59(1.82,3.36) | 3.35(2.6,4.14) | 3.3(2.53,4.07) |

All three tables show that the predictions of Z-Generation subjects were less accurate than those for people born before 1990, due to the original diversity of this group in the sample data.

As it can be seen, approximately, predictions of the competence score for the five areas varied in the range 1–3 for the minimum ETS individuals, in the range 1.5–3.4 for medium ETS ones, and in between 2 and 4 for maximum ETS individuals.

## 5. Discussion

### 5.1. Global Digital Competence

The assessment of digital teaching competence was conducted according to the DigCompEdu framework, which is the one most highly valued by experts [41]. With respect to the first goal, based on the global digital competence (GDC), as assessed by the grand mean of the items in the questionnaire used in this study, the gender divide turned out to be evident. Global digital competence scores were consistent with the self-perception of subjects on their own tech skill development on common issues—those that perceived themselves as less prepared for tech were those who got smaller level scores in digital competence as evaluated by the quest, and vice versa. However, the lack of differences among branches of knowledge would lead us to claim that the university system in Spain contributes homogeneously, in mean terms, to the digital training of undergraduate students. This would endorse the proposal of a standard baseline to begin digital training in Masters of Education to train teachers whatever their academic origin, but considering gender and basic digital skills differences.

Recall that the gender digital divide emerging in this study has also been shown in other related studies, with women scoring lower, like [15,20]. Nevertheless, it is convenient to remark that the global digital competence has been evaluated in terms of an average that dilutes possible differences in certain areas of competence. To further investigate digital competence, a distinction by area of competence has been considered and studied here.

### 5.2. Areas of Competence

When considering specific scores in the different digital competence areas, differences have been encountered related to area, gender, branch of knowledge, and age (Z-Generation). Without a doubt, this study gives more effective guidelines to attend the student population studying a Master's in Education in Spain to become teachers at the secondary level. Again, self-perception of individuals on everyday tech issues has been decisive in explaining the digital performance of individuals related to digital training for teaching.

### 5.3. Digital Competence by Area

Initial digital competence has been proven to be different for graduate students embarking on the Master's in Education, for the diverse areas of competence considered in DigComp, say: (A1) information and information literacy, (A2) communication and collaboration, (A3) digital content creation, (A4) security, and (A5) problem solving. Specifically, students proved to be more competent in competence area A1, the most related to everyday uses of technology, followed by areas A2, A4, A5, and finally area A3. This is consistent with conclusions reached by [11], but contradictory to those by [12].

Moreover, competence area development does not behave the same for individuals with different everyday tech skills (ETS). For example, distance between performance in consecutive areas A3 and A5 was greater when the ETS increased: distance was 0.33 for min-ETS, 0.53 for med-ETS, and 0.7 for max-ETS. The area A5 score, on the contrary, was quite near the A4 score for max-ETS, but not for the other groups (med and min ETS).

Area A3 was revealed as the least developed competence area, however digital content generation is so crucial for teaching in the XXI century. So, specific training in this area should be promoted in the Master's in Education in Spain to adequately prepare competent teachers.

### 5.4. Digital Competence by Gender

Gender differences encountered when studying global digital competence, were confirmed again in all areas in a homogeneous way. Women showed a competence level 0.11 below Men in all areas, which confirmed a digital divide by gender. This result was consistent with the findings of [15,17–19] and partially with those of [42], since this coincidence was only shown in ICT knowledge, as well as

those in Reference [20] and [13]. However, it contradicts the conclusions of [16], and related to teachers, the conclusions of [14,21–25].

### 5.5. Digital Competence by Branch of Knowledge

The branch of knowledge for which a lower score was predicted for all the areas of competence was science (S), followed by social and legal sciences (SLS) and then arts and humanities (AH). The one that scored the highest was engineering and architecture (EA). The literature consulted, [23], agreed in highlighting that the teachers from technological degrees have greater digital competence.

### 5.6. Digital Competence by Age

Individuals born from 1990 (Z-Generation), scored higher in all competence areas. However, this effect was not the same for each group in everyday tech skills. Subjects born since 1990 that perceived themselves with a minimum level of technological skills in everyday matters scored significantly higher than those born before 1990 in all competence areas. This fact did not occur for subjects who declared themselves on a medium scale of technological skills; in this group, those born before and after 1990 scored almost equally (on average) in all competence areas. Finally, in the group of subjects who recognized themselves at a maximum level of technological skills in everyday matters, those of generation Z were again separated (in their competence scores) from those who were born before earlier, but lesser than in the medium group.

These results are consistent with other research developed on the subject, with samples of students [28,29], and with samples of teachers [21,24,25,33]. However, they are at odds with the conclusions reached by others [23,26,30–32].

## 6. Conclusions

This study has made it possible to conclude, on the one hand, on the homogeneity of the university degrees of the Bologna Plan (from students entering MasterProf) with respect to mean digital training. On the other hand, more exhaustively, differences have come to light concerning specific training on the different digital competence areas in DigComp related to gender, branch of knowledge, age, and by taking into account the self-perception of individuals on their own capabilities in everyday technological issues.

This last variable, self-perception of individuals on everyday technological matters, has been instrumental in explaining variability that the other variables were not able to explain. Moreover, digital evaluation has proved to be consistent with previous self-perception of individuals on their own abilities with the use of ICT resources.

As derived from the study, it is necessary to continue working on digital competence training, especially for the group of graduates who aspire to become teachers, also by promoting females, older people, and nontechnological degrees. Digital training in critical competence areas for teaching, such as the creation of digital content, suffers from major shortcomings for all types of graduates embarking on a Master's degree in Education.

The need and effects of incorporating an ICT syllabus into the subjects of a Master's in Education will be perceived in some years, hence the need to continue investigating this issue over the coming years.

## 7. Limitations

This study was based on a university group in XXX (Spain) that could clearly be considered representative of the group made up of graduate students in Spain who aspire to become secondary school teachers in the coming years. However, some limitations could be identified.

This study was based on a self-perception questionnaire, and not on objective evaluations of digital competence. The objective nature of the scores in the dimensions considered was improved by using averages of several items that make up each dimension.

The tool for retrieving information was a simplified version of the DigComp questionnaire, that though validated through Cronbach indexes, is not guaranteed to provide identical results. However, formulation of questions is quite similar to formulation in the original quest.

The sample contained 40% of the students studying MasterProf in XXX during the 2017–2018, 2018–2019, and 2019–2020 academic years, so it is a representative sample for this population. Regarding sociological issues concerning gender, this sample was not so different from the students starting any master's degree in Spain—there are54.8% females in Masters in Spain and 61.4% in MasterProf.

**Author Contributions:** Conceptualization, V.G.-C., D.J.-H. and A.T.-S.; methodology, A.M.M. and V.G.-C.; software, A.M.M. and J.M.; formal analysis, A.M.M. and J.M.; investigation, A.M.M., J.M., V.G.-C., D.J.-H. and A.T.-S.; writing—original draft preparation, A.M.M., J.M., V.G.-C., D.J.-H. and A.T.-S.; writing—review and editing, A.M.M., J.M., V.G.-C., D.J.-H. and A.T.-S.; supervision, A.M.M. and V.G.-C. All authors have read and agreed to the published version of the manuscript.

**Funding:** This research received no external funding.

**Conflicts of Interest:** The authors declare no conflict of interest.

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
