# Peer review of "Digital Competence of Future Secondary School Teachers: Differences According to Gender, Age, and Branch of Knowledge"

_sustainability, doi:10.3390/su12229473_

Round 1

Reviewer 1 Report

Overall, the article is very good, clearly written and provides well-thought insights. However, I have few comments:

1) I don´t quite understand line 97, where you write " we find a number of jobs..." - the examples you provide are all about teachers. Did you mean to write "jobs"?

2) line 129 - "...day-to-day." Did you probably mean to write another word after this? 

3) On a few occasions, you write the word "relating", but it seems that "related" would be more appropriate

4) on lines 166 and 169, you talk about "quest" - did you mean questionnaire?

5) line 302 - "will necessarily scores" - did you mean "score"?

6) on lines 403-404, you say "The Branch of Knowledge for which a lower score is predicted for all the areas of competence is Science (S)" - I find this result surprising, as it is the opposite we have in my country. What may contribute to this result?

7) line 408 - "born from" is not an appropriate form

Author Response

1) I don´t quite understand line 97, where you write " we find a number of jobs..." - the examples you provide are all about teachers. Did you mean to write "jobs"?

‘Jobs’ has been changed to ‘studies’. (line 97)

2) line 129 - "...day-to-day." Did you probably mean to write another word after this? 

The word "use" has been added. (line 129)

3) On a few occasions, you write the word "relating", but it seems that "related" would be more appropriate

The work has been reviewed by a translation company, their recommendation being the word 'relating'. "Related to" only means that there is some kind of connection while "relating to" indicates something that is about the topic.

4) on lines 166 and 169, you talk about "quest" - did you mean questionnaire?

Change made.

5) line 302 - "will necessarily scores" - did you mean "score"?

Change made.

6) on lines 403-404, you say "The Branch of Knowledge for which a lower score is predicted for all the areas of competence is Science (S)" - I find this result surprising, as it is the opposite we have in my country. What may contribute to this result?

This work brings a knowledge that was not previously available, science students may have more knowledge of technologies, but not so much for teaching.

7) line 408 - "born from" is not an appropriate form

Change made.

Reviewer 2 Report

The problem chosen for the analysis in the article is relevant and significant. The topic is clearly worded, but the keywords are not harmonised with the topic. It is not clear why the words “21st century abilities” (line 25) are given here, because further this is not emphasized and substantiated. It is also not revealed how the concepts used in the article relate to each other, how they are related and what link with the main key concept – digital competency – exists. It should be worth thinking about the unification of concepts, which would allow a clearer understanding of the content of the article (e.g., line 26 - Information literacy; line 46 - technological skills;  line 51 - digital skills, and etc.)

The theoretical analysis clearly presented in the article is directly related to the topic of the article by distinguishing key variables that are related to the digital competence, but the applied research instrument encompasses more variables.

It is debatable whether the first aim can be empirically verified and revealed (lines 152-153). The statement that the percentage of students who took part in the survey allows to summarize the results for the whole population (lines 202-203) requires justification. On what basis is this stated?

I would recommend to more clearly define and to provide an explanation of how research participants were selected. What principles of research ethics were implemented in the study. It is necessary to discuss the ethical principles that were implemented while conducting the research.

The presented results are significant and relevant. Presentation of tables after the text without giving any subsequent analysis or generalization seems slightly strange (Tables 8, 9, 10).

In a general sense, paying attention to the obtained findings and the discussion, the article is interesting and enriches the understanding with new practices and ideas.

Author Response

Reviewer 2

The problem chosen for the analysis in the article is relevant and significant. The topic is clearly worded, but the keywords are not harmonised with the topic. It is not clear why the words “21st century abilities” (line 25) are given here, because further this is not emphasized and substantiated. It is also not revealed how the concepts used in the article relate to each other, how they are related and what link with the main key concept – digital competency – exists. It should be worth thinking about the unification of concepts, which would allow a clearer understanding of the content of the article (e.g., line 26 - Information literacy; line 46 - technological skills;  line 51 - digital skills, and etc.).

Concepts have been unified according to the reviewer's recommendation.

The theoretical analysis clearly presented in the article is directly related to the topic of the article by distinguishing key variables that are related to the digital competence, but the applied research instrument encompasses more variables.

In the space provided in the article it is not complicated to be able to mention all the variables. We have thought it necessary only to comment on those that have been included in the theoretical framework in order to understand the work in a global way.

It is debatable whether the first aim can be empirically verified and revealed (lines 152-153). The statement that the percentage of students who took part in the survey allows to summarize the results for the whole population (lines 202-203) requires justification. On what basis is this stated?

I would recommend to more clearly define and to provide an explanation of how research participants were selected.

All these elements have been clarified from line 205 to 207. “The available sample supposes 40.4% of the total enrolment for all three years of the MasterProf progamme, so it can be considered as representative sample of those who have completed the master's degree in question. The sample was selected for convenience, asking those enrolled in the Master's to answer the questionnaire anonymously.”

What principles of research ethics were implemented in the study. It is necessary to discuss the ethical principles that were implemented while conducting the research.

Incorporated in lines 159-161. The research was carried out following all the ethical principles established by the Office of Responsible Research of the University Miguel Hernández of Elche.

The presented results are significant and relevant. Presentation of tables after the text without giving any subsequent analysis or generalization seems slightly strange (Tables 8, 9, 10).

The tables reflect the results of the Bayesian analysis.

In a general sense, paying attention to the obtained findings and the discussion, the article is interesting and enriches the understanding with new practices and ideas.

Thank you very much for your words on the article.

Reviewer 3 Report

Congratulations to the authors. It is interesting to focus on the initial situation of digital competence of the teaching staff studying the master's degree with a view to proposing / modifying the study plan and incorporating training regarding ICT. This study not only detects a need, but also supports a relevant change at the curriculum level. The 3 factors to be analyzed have been extensively reviewed at a theoretical level, supporting the conduct of this study. Materials and methods are correctly defined. The results are clear and well structured. It is interesting to observe the similar starting point between degrees, such as the detection of the need for ICT training during the master's degree. I believe that it is a study that addresses a subject with a long history in recent years, but whose conclusions provide answers for decision-making and the improvement of teacher training regarding digital competence. Best regards

Author Response

Thank you very much for your words on the article.